# The Prognosis Performance of a Neutrophil- and Lymphocyte-Associated Gene Mutation Score in a Head and Neck Cancer Cohort

**DOI:** 10.3390/biomedicines11123113

**Published:** 2023-11-22

**Authors:** Tsung-Jang Yeh, Hui-Ching Wang, Shih-Feng Cho, Chun-Chieh Wu, Tzu-Yu Hsieh, Chien-Tzu Huang, Min-Hong Wang, Tzer-Ming Chuang, Yuh-Ching Gau, Jeng-Shiun Du, Yi-Chang Liu, Hui-Hua Hsiao, Mei-Ren Pan, Li-Tzong Chen, Sin-Hua Moi

**Affiliations:** 1Division of Hematology & Oncology, Department of Internal Medicine, Kaohsiung Medical University Hospital, Kaohsiung Medical University, Kaohsiung 807, Taiwan; aw7719@gmail.com (T.-J.Y.); joellewang66@gmail.com (H.-C.W.); sifong96@gmail.com (S.-F.C.); cathyvioletevergarden@gmail.com (T.-Y.H.); gankay18@hotmail.com (C.-T.H.); dhlsy01128@gmail.com (M.-H.W.); benjer6@gmail.com (T.-M.C.); cheesecaketwin@gmail.com (Y.-C.G.); ashiun@gmail.com (J.-S.D.); ycliu@cc.kmu.edu.tw (Y.-C.L.); huhuhs@kmu.edu.tw (H.-H.H.); 2Graduate Institute of Clinical Medicine, College of Medicine, Kaohsiung Medical University, Kaohsiung 807, Taiwan; pan.meiren.0324@gmail.com; 3Center for Cancer Research, Kaohsiung Medical University, Kaohsiung 807, Taiwan; leochen@nhri.org.tw; 4Faculty of Medicine, College of Medicine, Kaohsiung Medical University, Kaohsiung 807, Taiwan; 5Department of Pathology, Kaohsiung Medical University Hospital, Kaohsiung Medical University, Kaohsiung 807, Taiwan; lazzz.wu@gmail.com; 6Drug Development and Value Creation Research Center, Kaohsiung Medical University, Kaohsiung 807, Taiwan; 7Department of Medical Research, Kaohsiung Medical University Hospital, Kaohsiung Medical University, Kaohsiung 807, Taiwan; 8Division of Gastroenterology, Department of Internal Medicine, Kaohsiung Medical University Hospital, Kaohsiung Medical University, Kaohsiung 807, Taiwan; 9National Institute of Cancer Research, National Health Research Institutes, Tainan 704, Taiwan; 10Research Center for Precision Environmental Medicine, Kaohsiung Medical University, Kaohsiung 807, Taiwan

**Keywords:** genomic mutation signature, neutrophil to lymphocyte ratio, head and neck cancer, prognostic biomarker

## Abstract

The treatment of head and neck squamous cell carcinomas (HNSCCs) is multimodal, and chemoradiotherapy (CRT) is a critical component. However, the availability of predictive or prognostic markers in patients with HNSCC is limited. Inflammation is a well-documented factor in cancer, and several parameters have been studied, with the neutrophil-to-lymphocyte ratio (NLR) being the most promising. The NLR is the most extensively researched clinical biomarker in various solid tumors, including HNSCC. In our study, we collected clinical and next-generation sequencing (NGS) data with targeted sequencing information from 107 patients with HNSCC who underwent CRT. The difference in the NLR between the good response group and the poor response group was significant, with more patients having a high NLR in the poor response group. We also examined the genetic alterations linked to the NLR and found a total of 41 associated genes across eight common pathways searched from the KEGG database. The overall mutation rate was low, and there was no significant mutation difference between the low- and high-NLR groups. Using a multivariate binomial generalized linear model, we identified three candidate genes (*MAP2K2*, *MAP2K4*, and *ABL1*) that showed significant results and were used to create a gene mutation score (GMS). Using the NLR-GMS category, we noticed that the high-NLR-GMS group had significantly shorter relapse-free survival compared to the intermediate- or low-NLR-GMS groups.

## 1. Introduction

Head and neck squamous cell carcinoma (HNSCC) is mostly derived from the mucosal epithelium of the oral cavity, pharynx, and larynx [1]. It is the most common malignancy of the upper aerodigestive tract and was ranked fourth in a Taiwanese male cohort study [2,3]. HNSCC treatment is generally multimodal and differs according to the disease stage, anatomical location, and surgical accessibility to achieve the most curative approach while optimizing the preservation of function [1,4]. Some early-stage diseases are curable with surgery or definitive radiotherapy; however, more than 60% of patients present with locally advanced disease upon diagnosis [5].

Concurrent platinum-based chemotherapy and radiotherapy, also referred to as chemoradiotherapy (CRT) or concurrent chemoradiotherapy (CCRT), plays an important role in HNSCC treatment. When considering organ preservation, definite CRT is recommended as a nonsurgical treatment for most patients with advanced pharyngeal and laryngeal cancers [6]. In postoperative management, the efficacy of adjuvant CRT has been proven in two multicenter randomized trials (EORTC 22931 and RTOG 9501) for high-risk patients with HNSCC, especially those with extranodal extension or positive surgical margins [7,8]. Despite the development of risk-adapted curative treatment strategies and other progress in therapeutic modalities, the overall 5-year survival is only 50%, and 65% of patients with an advanced stage of the disease have significantly compromised survival [9]. Therefore, in addition to the development of novel treatment approaches, the search for predictive or prognostic markers in patients with HNSCC is necessary.

To date, approximately 70 markers have been evaluated and reported from either blood or tumor tissues [10,11,12]. In the conventional treatment era, epidermal growth factor receptor (EGFR), p16, human papillomavirus (HPV), cyclin D1 (CCND1), B cell lymphoma-extra large (Bcl-xL)/Bcl-2 and excision repair cross complementation group 1 (ERCC1) have been identified as possible prognostic markers in clinical trials [11,12]. In the current immunotherapy era, programmed death ligand-1 (PD-L1) expression, tumor mutational burden (TMB), microsatellite instability (MSI), HPV status, smoking status, circulating tumor cells (CTCs), circulating tumor DNA (ctDNA), gut or oral cavity microbiota, and tumor-microenvironment-related gene expression profiles have been suggested as potential immune biomarkers to predict the efficacy of immune checkpoint inhibitors [12]. 

In 1863, Virchow established a connection between inflammation and cancer based on his observations [13]. Since then, research on the association between inflammation and carcinogenesis has increased, supporting Virchow’s theory [14]. Inflammation is now characterized as a critical component of tumor progression based on its contribution to the multiple hallmarks of tumorigenesis [15,16,17,18]. Several inflammatory parameters have been reported, such as C-reactive protein, the neutrophil-to-lymphocyte ratio (NLR), the platelet-to-lymphocyte ratio (PLR), and the lymphocyte-to-monocyte ratio (LMR) [19,20,21,22,23,24]. Among these, the NLR is the most studied and promising clinical biomarker and has been shown to be prognostic in many solid tumors, including HNSCC [10,25,26,27,28,29,30,31,32,33]. However, there are still many unknown areas to explore, and research on inflammatory biomarkers and cancer genomic mutations is lacking. This study aimed to decipher the predictive value of the NLR in patients with locally advanced HNSCC treated with CRT and to explore the associations between the NLR and the cancer genomic landscape.

## 2. Materials and Methods

### 2.1. Data Source

In this retrospective cohort study, all data were collected via the health information system of Kaohsiung Medical University Hospital under an approved protocol (KMUHIRB-E(I)-20210401). Eligible patients with histologically proven HNSCC (grades 1 to 3) originating in the oral cavity (OC), oropharynx (OPC), hypopharynx (HPC), or larynx (LC) were recruited between 2016 and 2022 at Kaohsiung Medical University Hospital, Taiwan. The Head and Neck Cancer Committee confirmed the tumor stage according to the 8th edition of the American Joint Committee on Cancer (AJCC)’s staging system. 

All patients underwent CRT, including 75 who received postoperative adjuvant therapy and 32 who received initial definitive treatment. CRT treatment included a total radiotherapy dose of 60–70 Gy and cisplatin-based chemotherapy. Other clinical data included detailed information on patient age, sex, tumor location, histological grade, clinical staging, body weight before and after CRT, changes in body mass index (BMI), and laboratory findings. 

### 2.2. Treatment Response

All patients were followed up with regularly at the Medical Oncology and Otorhinolaryngology outpatient departments. Disease status evaluation included tumor site inspection, laboratory examinations, and imaging studies. Treatment response was assessed and determined using computed tomography or magnetic resonance imaging at baseline and at three- to six-month intervals after treatment initiation. The treatment response of the patients was evaluated using Response Evaluation Criteria in Solid Tumors (RECIST) 1.1-measurable lesions and classified into four categories: complete response (CR), partial response (PR), stable disease (SD), and progressive disease (PD). CR and PR were classified as good responses, whereas SD and PD were classified as poor responses. The median follow-up duration in this cohort was 16.6 (range 2.2–80.9) months. 

### 2.3. Neutrophil-to-Lymphocyte Ratio (NLR)

The NLR was calculated as the simple ratio between the neutrophil and lymphocyte counts measured in the peripheral blood. Absolute lymphocyte count (ALC) and absolute neutrophil count (ANC) data were retrieved from four weeks prior to commencing radiotherapy. If multiple values were available before treatment, those closest to the start date of radiotherapy were selected. The neutrophil-to-lymphocyte ratio (NLR) was computed using neutrophil and lymphocyte measurements and dichotomized into low- and high-NLR categories using a receiver operating characteristic (ROC) analysis.

### 2.4. Somatic Gene Mutation Profiles and Candidate Genes

The somatic gene mutation profiles of the study cohort were determined using next-generation sequencing (NGS) and FoundationOne CDx (F1CDx), according to the Illumina^®^ HiSeq 4000 platform, using formalin-fixed paraffin-embedded (FFPE) HNSCC tissue specimens. The F1CDx-targeted NGS platform method was validated previously [34]. Neutrophil- and lymphocyte-associated pathways were identified, and the genes involved in these pathways were retrieved from the Kyoto Encyclopedia of Genes and Genomes (KEGG) database. The results of the somatic gene mutation profiles in our study cohort were subsequently mapped. A total of 41 genes associated with lymphocyte and neutrophil signaling pathways were mapped from the somatic gene mutation profiles of the study cohort and considered as candidate gene panels for later analyses. The somatic mutation rates of the candidate genes in the study cohort and the NLR categories were summarized, and the difference between the NLR categories was estimated using Fisher’s exact or Pearson’s chi-squared test. 

### 2.5. Gene Mutation Score (GMS)

A multivariate binomial generalized linear model was used to evaluate the association between the NLR and somatic mutations in the candidate genes. Candidate genes with significant results derived from the multivariate model were further selected to generate the GMS, which was calculated by multiplying the estimated coefficient in the multivariate model by the gene mutation status (wild as 0, mutated as 1). Subsequently, the study cohort was dichotomized into low- and high-GMS categories using an ROC analysis. NLR-GMS categories were generated using both the NLR and GMS, and the study cohort was reassigned into low-, intermediate-, and high-NLR-GMS categories based on their NLR and GMS categories. Patients with both a low NLR and GMS were categorized as having a low NLR-GMS, those with both a high NLR and GMS were categorized as having a high NLR-GMS, and the remaining patients were considered to have an intermediate NLR-GMS.

### 2.6. Statistical Analysis

The baseline characteristics of the study cohort were summarized using frequencies and percentages, and laboratory measurements were summarized using medians and interquartile ranges. Differences in baseline characteristics and laboratory measurements between the good- and poor-response subgroups were estimated using Fisher’s exact test, Pearson’s chi-squared test, or the Wilcoxon rank-sum test. The predictive performances of both the NLR and NLR-GMS for treatment response and short-term relapse-free survival within 36 months were evaluated using an ROC analysis. The area under the ROC curve (AUC) was used to determine the predictive performance of both the NLR and NLR-GMS for progression-free survival (PFS). A higher AUC indicated a better predictive performance. The survival rate of each NLR and NLR-GMS category was estimated using the Kaplan–Meier estimator, and the survival difference between subcategories was estimated using the log-rank test. All *p*-values were two-sided, and *p* < 0.05 was considered statistically significant. All analyses were performed using R 4.1.2 software (R Core Team, 2021, Vienna, Austria).

### 2.7. Immunohistochemistry

The HNSCC specimens were fixed on paraffin-embedded biopsies and sectioned. The slices were deparaffinized using xylene and then dehydrated with ethanol. Endogenous peroxidase activity was quenched with 3% hydrogen peroxide containing methanol for 15 min. The sections were heated in 100 mmol/L citrate buffer for 10 min to revive the antigens. The tissues were incubated with 3 primary antibodies at room temperature for 30 min and then rinsed three times with phosphate-buffered saline (PBS) according to the manufacturer’s protocol. Following color development, we applied cover slips to the sections and observed them under a microscope. Staining intensity in the cancer tissue was independently examined by pathologists who were blinded to the patients’ clinical features and outcomes. The following primary antibodies were used: anti-ABL1 (1:100, Elabscience, Houston, TX, USA), anti-MAP2K2 (1:100, Elabscience, Houston, TX, USA), and anti-MAP2K4 (1:100, Elabscience, State of Texas, USA). In the assessment of IHC staining, staining intensity ranged from 0 (negative) to 3+ (high strength) with the percentage of positively labeled cells.

## 3. Results

### 3.1. Baseline Characteristics of Patients

Between 2016 and 2022, 107 patients were enrolled in this study. All patients were diagnosed with HNSCC and underwent CRT. Patient characteristics, including age, sex, tumor location, pathological grade, stage, pre-CRT BMI, post-CRT BMI, body weight loss, white blood cell count, ANC, ALC, and the NLR, are summarized in Table 1. The majority of the patients were middle-aged (65.4% between 45 and 64 years), male (93.5%), had oral cavity cancer (67.3%), had grade 2 disease (55.8%), and were stage IV (85.0%). Overall, 84 patients had a good response to CRT, and 23 patients had a poor response to CRT. 

There were no statistically significant differences between the good- and poor-response groups with regards to age, sex, pathological grade, clinical cancer stage, pre-CRT BMI, post-CRT BMI, body weight loss, white blood cell count, ANC, and ALC. However, there were significantly more oral cavity cancers in the poor-response group (*p* = 0.038). The difference in the NLR between the good-response group and poor-response group was also prominent (*p* = 0.037), with more patients having a high NLR (≥2.7) in the poor-response group (87.0% vs. 64.3%). 

### 3.2. Signaling Pathways Associated with Lymphocytes and Neutrophils

After confirming that a high NLR was related to a poor treatment response, we further analyzed the genetic alterations associated with lymphocytes and neutrophils. Table 2 summarizes eight common pathways associated with lymphocytes and neutrophils which were identified using the KEGG database, including phosphoinositide 3-kinase (PI3K) and Fc gamma receptor IIb (FcγRIIb) signaling in B lymphocytes, protein kinase C (PKC) and 4-1BB signaling in T lymphocytes, the regulation of IL-2 expression in activated and anergic T lymphocytes, cytotoxic T-lymphocyte antigen 4 (CTLA4) signaling in cytotoxic T lymphocytes, CD27 signaling in lymphocytes, and N-formyl methionyl-leucyl-phenylalanine (fMLP) signaling in neutrophils. 

Table 2 reports the pathways associated with the lymphocytes and neutrophils derived using the mutated genes detected in this cohort. The “involved genes” column indicates the mutated genes found in our study cohort, which are simultaneously involved in the corresponding pathways. The “genes in pathways” column reported the overall number of genes involved in the correspond pathways, and the “pathway percentage” was computed by dividing the “involved genes” column by the “genes in pathways” column, denoting the percentage of mutated genes found in this study cohort involving a correspondence pathway. The raw NGS data for the somatic mutation profiles of this study cohort are in Appendix A. 

### 3.3. Somatic Mutation Profiles of Candidate Genes

We matched the genes in these eight pathways with the somatic gene mutation profiles of the patients. A total of 41 genes were identified and were considered candidate gene panels for subsequent analyses, as shown in Table 3. Overall, gene mutation rates were low. Only nine genes had mutation rates of over 10%: *PIK3CA* (23.4%), *PRKCI* (18.7%), *CASP8* (18.7%), *PIK3C2G* (15.0%), *MAP3K13* (13.1%), *ATM* (11.2%), *JAK2* (11.2%), *PTEN* (10.3%), and *CARD11* (10.3%). Among these 41 genes, there were no mutation differences between the low-NLR group and the high-NLR group. 

### 3.4. Predictive Performance of GMS and NLR-GMS

Due to the relatively low mutation rate, we used a multivariate binomial generalized linear model to evaluate the association between the NLR and somatic mutations of the candidate genes (Table 4). Three candidate genes (*MAP2K2*, *MAP2K4,* and *ABL1*) exhibiting significant results derived from the multivariate model were selected to generate the GMS. *MAP2K2* mutations were associated with a low NLR, whereas *ABL1* and *MAP2K4* mutations were associated with a high NLR. 

NLR-GMS categories were generated using both the NLR and GMS, and the study cohort was reassigned into low-, intermediate-, and high-NLR-GMS categories based on their NLR and GMS categories. Figure 1A,B display ROC plots for the NLR category and the NLR-GMS for treatment response and relapse events within 36 months, respectively. Both exhibited improved predictive performance using the NLR-GMS categories compared to the NLR categories. 

Although the high-NLR group exhibited poorer survival than the low-NLR group, no significant survival difference was found between the groups (Figure 1C). However, when we used the NLR-GMS categories, the difference in relapse-free survival between the NLR-GMS groups was significant (overall *p* = 0.003; Figure 1D). The high NLR-GMS group had a significantly shorter relapse-free survival period than the intermediate (*p* = 0.004) and low (*p* = 0.002) NLR-GMS groups. 

### 3.5. Somatic Mutation Validation via Immunohistochemistry Staining

Based on the result of the multivariate binomial generalized linear model, three candidate genes (*MAP2K2*, *MAP2K4* and *ABL1*) exhibited significant results and were used to generate the GMS. We further validated the expression of three candidate genes by immunohistochemistry (IHC) staining cancer tissues from HNSCC patients. Figure 2 shows representative images of IHC staining for the mutated and wild type of each candidate gene. The yellow words in the upper-left corner of each picture indicate the staining intensity and the percentage of positively labeled cells. Cancer tissues with somatic mutations of the three candidate genes showed a higher staining intensity and percentage of positive cells compared to wild-type specimens. Overall, it showed an increasing IHC intensity in somatic mutated samples.

## 4. Discussion

The relationship between inflammation and cancer is highly associated. Inflammation predisposes patients to cancer development and promotes all stages of tumorigenesis [15,16,17,18]. Several acquired factors are already proven to be carcinogenic and are often associated with chronic inflammation, including chronic bacterial and viral infections, autoimmune diseases, environmental factors (asbestos exposure), lifestyle factors (obesity, tobacco smoking, and excessive alcohol consumption), and aging, which are thought to promote tumor-extrinsic inflammation [14,35]. In contrast, tumor-intrinsic inflammation, also referred to as cancer-elicited inflammation, is induced after tumor initiation and contributes to malignant progression by recruiting and activating inflammatory cells [14,17,36]. Moreover, anticancer therapies can also induce inflammation via the necrosis and necroptosis of cancer cells [14,17]. Overall, cancer cells and surrounding stromal and inflammatory cells engage in well-orchestrated reciprocal interactions to form an inflammatory tumor microenvironment [17].

Several parameters have been reported to represent the degree of systemic inflammation, such as C-reactive protein and the NLR, PLR, and LMR [19,20,21,22,23,24]. The NLR is the most promising clinical biomarker and has been shown to be prognostic in many solid tumors, including HNSCC [25,26,27,28,29,30,31,32,33]. The NLR reflects the dynamic relationship between neutrophils (innate immune response) and lymphocytes (adaptive immune response). During stress, trauma, surgery, systemic infection, inflammation, sepsis, or critical illness, the dysregulation of innate and adaptive immune responses results in neutrophilia and lymphocytopenia [37]. Initially, the NLR was used as an index of systemic inflammatory response syndrome (SIRS) and stress in critically ill patients [38]. However, more recently, the NLR has been applied in almost all medical scenarios as a reliable and easily available marker of immune response to various stimuli [37].

The NLR was first applied as a prognostic factor for colorectal cancer in 2005 by Walsh et al. [39]. Two meta-analyses [25,26] and various studies have demonstrated that the NLR is a prognostic factor in different solid tumors, including esophageal cancer [40], gastric cancer [41,42], pancreatic cancer [43], and biliary tract cancer [44], etc. The role of the NLR in HNSCC was also evaluated in many studies, and all of them concluded that the NLR is a reliable prognostic marker [28,29,30,33,45,46,47,48,49,50,51,52,53,54]. Similarly, our cohort demonstrated patients with a high NLR in the poor-response group (87.0% vs. 64.3%), although there was no difference in the 36-month relapse-free survival between the high- and low-NLR groups.

To further evaluate the genetic alterations of the immune response associated with the NLR, we analyzed the NGS data of HNSCC patients and matched them with the KEGG database. Eight pathways and forty-one associated genes were identified. Different statistical methods were applied to determine the association between the NLR and somatic mutations in these candidate genes. Three candidate genes (*MAP2K2*, *MAP2K4*, and *ABL1*) were selected using a multivariate binomial generalized linear model and were further advanced to GMS generation.

*MAP2K2* and *MAP2K4* both belong to the mitogen-activated protein kinase (MAPK) family. *MAP2K2* regulates the phosphorylation and activation of extracellular signal-regulated kinases (ERKs) [55]. *MAP2K4* can activate p38 via the phosphorylation of c-Jun-N-terminal kinases (JNK) [56]. MAPK cascades regulate a wide variety of cellular processes including proliferation, differentiation, transcriptional regulation, and stress responses [55,57]. MAPK activation also plays critical roles in the production of pro-inflammatory cytokines and the induction of the expression of multiple inflammatory- associated genes [58,59,60,61,62]. A variety of pharmacological inhibitors have been developed to specifically block MAPK kinase 1 and 2 (MEK1/2) [63,64]. Four MEK inhibitors, Trametinib, Cobimetinib, Binimetinib, and Selumetinib, have already been approved by the FDA to date [65]. An emerging technology, named proteolysis targeting chimera (PROTAC), could break the limitation of acquired resistance during long-term treatment by inducing MEK1/2 degradation [65]. Unlike *MAP2K2*, inhibitors specifically targeting *MAP2K4* are few. PLX8725, a novel *MAP2K4* inhibitor, demonstrates promising in vivo activity against patient-derived xenografts of uterine leiomyosarcomas harboring gain-of-function alterations in the *MAP2K4* gene [66]. In fact, *MAP2K4* mutations are sensitive to MEK inhibitors in multiple cancer models [67]. There are also a variety of potent inhibitors of the p38 MAP kinases have been developed, such as SB203580, SB202190, and BIRB-796 [64].

*ABL1* is a proto-oncogene that encodes a protein tyrosine kinase involved in a variety of cellular processes, including cell division, adhesion, differentiation, and response to stress. *ABL1* is involved in the occurrence and development of several types of cancers including colon, kidney, and breast cancer [68,69]. Some inflammatory conditions are also associated with *ABL1* [70]. *ABL1* mediates inflammation by regulating the NF-κB and STAT3 signaling pathways. The blockade of *ABL1* suppresses inflammatory signaling and cytokines [71]. To date, there are several approved *ABL1* inhibitors, such as imatinib, nilotinib, dasatinib, bosutinib, ponatinib, and so on. Most of them are treatments for chronic myeloid leukemia, which is *BCR-ABL1*-positive.

In short, three candidate genes (*MAP2K2*, *MAP2K4*, and *ABL1*) are all associated with inflammation and carcinogenesis. In our study, by using the NLR-GMS categories to divide patients into three groups, we noticed that the high-NLR-GMS group had a significantly shorter relapse-free survival than the intermediate- and low-NLR-GMS groups. In conclusion, the GMS could further target extremely high-risk patients with worse short-term survival based on the NLR-GMS categories.

This article possesses some strengths and limitations. Initially, the previous publications only affirmed the association between the NLR and treatment response. However, in this study, we conducted a comprehensive genetic investigation along with an investigation of clinical biomarkers in patients with HNSCC who underwent CRT and devised NLR-GMS categories to predict survival. Nonetheless, this cohort has a relatively small sample size, and there were differences in the tumor locations between the two groups. Specifically, the poor-response group had more oral cavity cancers, while the good-response group had more oropharynx cancers. Since different anatomical sites represent distinct etiological factors and background genetic alterations, this may have affected the study results. The longitudinal pattern (or variation) in the NLR of the study population was not investigated due to the nature of the study’s design. Additionally, further experiments are necessary to elucidate the role, influence, and mechanism of the candidate genes (*MAP2K2*, *MAP2K4*, and *ABL1*), which could serve as a valuable theme for future research.

## 5. Conclusions

In our cohort of HNSCC patients that underwent CRT, a high pre-treatment NLR is linked to a poor response. We conducted a further analysis on the genetic alterations associated with the NLR using somatic gene mutation profiles and identified 41 associated genes. However, there was no significant difference observed between the high- and low-NLR groups. Through the use of a multivariate binomial generalized linear model, we selected *MAP2K2*, *MAP2K4*, and *ABL1* to develop a GMS. By utilizing the NLR-GMS categories, we could effectively target high-risk patients with an extremely poor short-term survival outcome.

## Figures and Tables

**Figure 1 biomedicines-11-03113-f001:**
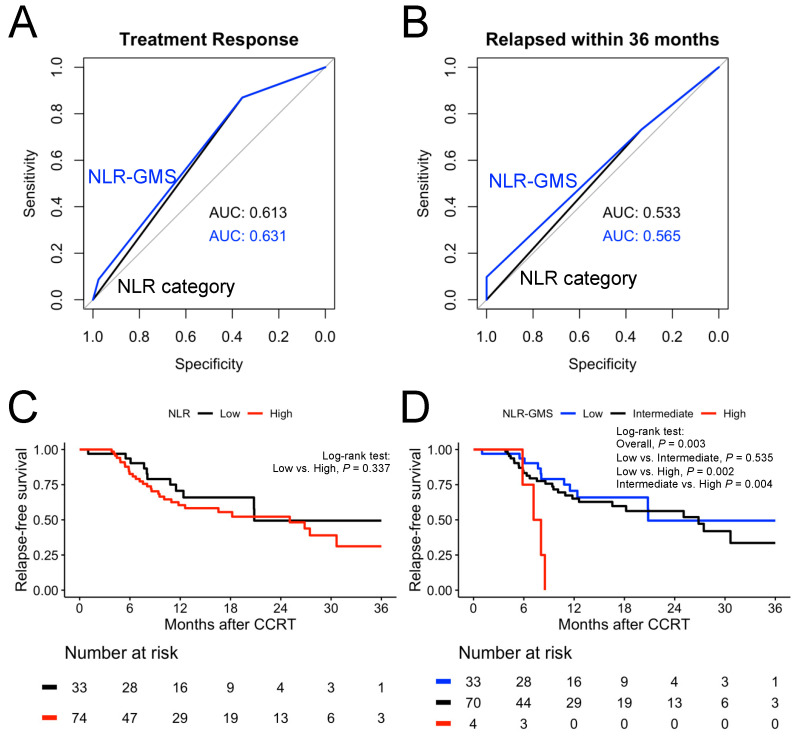
Predictive performance of NLR and NLR-GMS in treatment response and short-term relapse-free survival (within 36 months). ROC plots for NLR category and NLR-GMS for (**A**) treatment response and (**B**) relapse event within 36 months. Kaplan–Meier plot for short-term relapse-free survival comparison according to (**C**) NLR and (**D**) NLR-GMS category. ROC, receiver operating characteristics. AUC, area under ROC curve.

**Figure 2 biomedicines-11-03113-f002:**
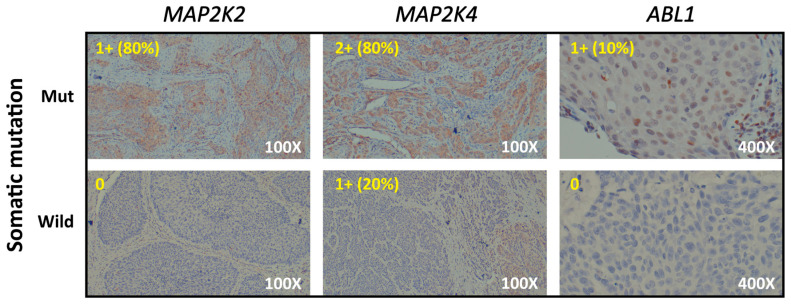
Representative immunohistochemistry (IHC) images of *MAP2K2*, *MAP2K4*, and *ABL1* expression from cancer tissues from HNSCC patients.

**Table 1 biomedicines-11-03113-t001:** Baseline characteristics of HNSCC cohort.

Characteristics	Overall, *n* = 107	Good Response, *n* = 84	Poor Response, *n* = 23	*p*
Age				0.804
<45	8 (7.5%)	6 (7.1%)	2 (8.7%)	
>65	29 (27.1%)	22 (26.2%)	7 (30.4%)	
45–64	70 (65.4%)	56 (66.7%)	14 (60.9%)	
Sex				0.168
Female	7 (6.5%)	4 (4.8%)	3 (13.0%)	
Male	100 (93.5%)	80 (95.2%)	20 (87.0%)	
Location				**0.038**
Hypopharynx	12 (11.2%)	12 (14.3%)	0 (0.0%)	
Larynx	2 (1.9%)	2 (2.4%)	0 (0.0%)	
Oral cavity	72 (67.3%)	51 (60.7%)	21 (91.3%)	
Oropharyx	21 (19.6%)	19 (22.6%)	2 (8.7%)	
Grade				0.617
Grade 1	28 (26.9%)	20 (24.7%)	8 (34.8%)	
Grade 2	58 (55.8%)	46 (56.8%)	12 (52.2%)	
Grade 3	18 (17.3%)	15 (18.5%)	3 (13.0%)	
Unknown	3 (2.8%)	3 (3.6%)	0 (0.0%)	
Stage				0.670
Stage I	3 (2.8%)	3 (3.6%)	0 (0.0%)	
Stage II	4 (3.7%)	4 (4.8%)	0 (0.0%)	
Stage III	9 (8.4%)	8 (9.5%)	1 (4.3%)	
Stage IV	91 (85.0%)	69 (82.1%)	22 (95.7%)	
BMI (Pre-CRT)	23.2 (14.6–34.1)	23.2 (15.1–34.0)	23.2 (14.6–34.1)	0.601
BMI (Post-CRT)	22.1 (13.9–33.6)	22.1 (13.9–33.6)	21.9 (14.5–33.0)	0.900
Body weight loss	−2.3 (−18.5–7.2)	−2.0 (−18.5–7.2)	−2.8 (−13.7–2.5)	0.377
White blood cell (/μL)	6810 (3020–35,150)	6660 (3020–15,170)	6970 (3710–35,150)	0.585
Neutrophils (Neu) (/μL)	68.8 (33.2–96.1)	68.4 (33.2–88.4)	70.0 (50.4–96.1)	0.147
Lymphocytes (Lym) (/μL)	20.8 (1.0–53.6)	21.0 (2.0–53.6)	20.7 (1.0–29.9)	0.147
NLR (Neu/Lym)				**0.037**
Low (<2.7)	33 (30.8%)	30 (35.7%)	3 (13.0%)	
High (≥2.7)	74 (69.2%)	54 (64.3%)	20 (87.0%)	

Abbreviation: BMI, body mass index; CRT, chemoradiotherapy; NLR, neutrophil-to-lymphocyte ratio.

**Table 2 biomedicines-11-03113-t002:** Pathways associated with lymphocytes and neutrophils derived using mutated genes in the HNSCC cohort.

Involved Pathway	Genes in Pathway	Involved Genes	Pathway Percentage	Gene Symbol
PI3K signaling in B lymphocytes	122	25	20.5	*LYN;IKBKE;AKT2;PIK3CA;AKT1;CD79A;IRS2;MAP2K2;PRKCI;CD79B;PTEN;NFKBIA;CBL; MAPK1;ABL1;PIK3CB;RAC1;SYK;HRAS;JUN;AKT3;MAP2K1;RAF1;KRAS;PIK3R1*
FcγRIIb signaling in B lymphocytes	41	14	34.1	*LYN;PIK3C2G;PIK3CA;AKT1;CD79A;ATM; CD79B;MAP2K4;PIK3CB;SYK;HRAS;PIK3C2B;KRAS;PIK3R1*
Regulation of IL-2 expression in activated and anergic T lymphocytes	75	16	21.3	*IKBKE;TGFBR2;SMAD2;CARD11;MAP2K2; NFKBIA;MAP2K4;MAPK1;RAC1;HRAS; MAP3K1;JUN;MAP2K1;RAF1;KRAS;SMAD4*
PKC signaling in T lymphocytes	107	17	15.9	*MAP3K13;PIK3C2G;IKBKE;CARD11;PIK3CA; ATM;NFKBIA;MAP2K4;MAPK1;PIK3CB;RAC1; HRAS;MAP3K1;PIK3C2B;JUN;KRAS; PIK3R1*
fMLP signaling in neutrophils	106	16	15.1	*GNAS;PIK3C2G;PIK3CA;MAP2K2;ATM;PRKCI;NFKBIA;MAPK1;PIK3CB;RAC1;HRAS; PIK3C2B;MAP2K1;RAF1;KRAS;PIK3R1*
CTLA4 signaling in cytotoxic T lymphocytes	82	13	15.9	*JAK2;PIK3C2G;AKT2;PIK3CA;AKT1;ATM; PIK3CB;SYK;PIK3C2B;AKT3;PTPN11;PPP2R2A; PIK3R1*
CD27 signaling in lymphocytes	51	10	19.6	*CASP8;MAP3K13;IKBKE;MAP2K2;NFKBIA; MAP2K4;MAP3K1;JUN;MAP2K1;BCL2L1*
4-IBB signaling in T lymphocytes	31	7	22.6	*IKBKE;MAP2K2;NFKBIA;MAP2K4;MAPK1; JUN;MAP2K1*

Abbreviation: PI3K, phosphoinositide 3-kinase; FcγRIIb, Fc gamma receptor IIb; PKC, protein kinase C; fMLP, N-formyl methionyl-leucyl-phenylalanine; CTLA4, cytotoxic T-lymphocyte antigen 4.

**Table 3 biomedicines-11-03113-t003:** Gene mutation rate of 41 genes associated with lymphocyte and neutrophil signaling pathways.

Genes	Overall, *n* = 107	Low NLR, *n* = 33	High NLR, *n* = 74	*p* Value
*PIK3CA*	25 (23.4%)	10 (30.3%)	15 (20.3%)	0.257
*PRKCI*	20 (18.7%)	7 (21.2%)	13 (17.6%)	0.655
*CASP8*	20 (18.7%)	3 (9.1%)	17 (23.0%)	0.089
*PIK3C2G*	16 (15.0%)	4 (12.1%)	12 (16.2%)	0.771
*MAP3K13*	14 (13.1%)	5 (15.2%)	9 (12.2%)	0.758
*ATM*	12 (11.2%)	5 (15.2%)	7 (9.5%)	0.508
*JAK2*	12 (11.2%)	3 (9.1%)	9 (12.2%)	0.751
*PTEN*	11 (10.3%)	4 (12.1%)	7 (9.5%)	0.735
*CARD11*	11 (10.3%)	1 (3.0%)	10 (13.5%)	0.167
*GNAS*	9 (8.4%)	2 (6.1%)	7 (9.5%)	0.718
*HRAS*	8 (7.5%)	3 (9.1%)	5 (6.8%)	0.700
*CBL*	7 (6.5%)	1 (3.0%)	6 (8.1%)	0.433
*IRS2*	6 (5.6%)	4 (12.1%)	2 (2.7%)	0.071
*MAP2K2*	6 (5.6%)	4 (12.1%)	2 (2.7%)	0.071
*MAP3K1*	6 (5.6%)	0 (0.0%)	6 (8.1%)	0.174
*NFKBIA*	5 (4.7%)	2 (6.1%)	3 (4.1%)	0.643
*PIK3CB*	5 (4.7%)	2 (6.1%)	3 (4.1%)	0.643
*KRAS*	5 (4.7%)	1 (3.0%)	4 (5.4%)	1.000
*TGFBR2*	5 (4.7%)	2 (6.1%)	3 (4.1%)	0.643
*LYN*	4 (3.7%)	1 (3.0%)	3 (4.1%)	1.000
*IKBKE*	3 (2.8%)	1 (3.0%)	2 (2.7%)	1.000
*AKT1*	3 (2.8%)	2 (6.1%)	1 (1.4%)	0.224
*CD79B*	3 (2.8%)	1 (3.0%)	2 (2.7%)	1.000
*MAPK1*	3 (2.8%)	2 (6.1%)	1 (1.4%)	0.224
*JUN*	3 (2.8%)	0 (0.0%)	3 (4.1%)	0.551
*MAP2K4*	3 (2.8%)	0 (0.0%)	3 (4.1%)	0.551
*PIK3C2B*	3 (2.8%)	2 (6.1%)	1 (1.4%)	0.224
*ABL1*	2 (1.9%)	0 (0.0%)	2 (2.7%)	1.000
*RAC1*	2 (1.9%)	0 (0.0%)	2 (2.7%)	1.000
*SYK*	2 (1.9%)	1 (3.0%)	1 (1.4%)	0.524
*AKT3*	2 (1.9%)	0 (0.0%)	2 (2.7%)	1.000
*RAF1*	2 (1.9%)	1 (3.0%)	1 (1.4%)	0.524
*PIK3R1*	2 (1.9%)	0 (0.0%)	2 (2.7%)	1.000
*SMAD4*	2 (1.9%)	2 (6.1%)	0 (0.0%)	0.093
*PTPN11*	2 (1.9%)	0 (0.0%)	2 (2.7%)	1.000
*BCL2L1*	2 (1.9%)	0 (0.0%)	2 (2.7%)	1.000
*AKT2*	1 (0.9%)	1 (3.0%)	0 (0.0%)	0.308
*CD79A*	1 (0.9%)	0 (0.0%)	1 (1.4%)	1.000
*MAP2K1*	1 (0.9%)	0 (0.0%)	1 (1.4%)	1.000
*SMAD2*	1 (0.9%)	1 (3.0%)	0 (0.0%)	0.308
*PPP2R2A*	1 (0.9%)	0 (0.0%)	1 (1.4%)	1.000

*p*-value is estimated using Fisher’s exact test or Pearson chi-squared test.

**Table 4 biomedicines-11-03113-t004:** Binomial generalized linear model result for the association between the NLR category and the mutation status of 41 associated genes.

Genes	Coefficients	SE	t	*p*
*LYN*	−0.386	0.336	−1.150	0.254
*IKBKE*	0.036	0.278	0.130	0.897
*AKT2*	−1.030	0.560	−1.842	0.070
*PIK3CA*	−0.486	0.251	−1.942	0.057
*AKT1*	−0.194	0.422	−0.461	0.646
*CD79A*	0.441	0.871	0.507	0.614
*IRS2*	−0.367	0.244	−1.506	0.137
* **MAP2K2** *	−0.749	0.315	−2.373	**0.021**
*PRKCI*	0.342	0.277	1.233	0.222
*CD79B*	−0.510	0.421	−1.212	0.230
*PTEN*	0.001	0.211	0.006	0.996
*NFKBIA*	0.384	0.359	1.069	0.289
*CBL*	0.219	0.277	0.791	0.432
*MAPK1*	0.088	0.447	0.196	0.845
* **ABL1** *	0.986	0.449	2.197	**0.032**
*PIK3CB*	0.149	0.298	0.501	0.618
*RAC1*	0.602	0.619	0.973	0.334
*SYK*	0.130	0.418	0.311	0.757
*HRAS*	−0.210	0.223	−0.942	0.350
*JUN*	0.698	0.466	1.500	0.138
*AKT3*	0.102	0.530	0.193	0.848
*MAP2K1*	−0.101	0.521	−0.193	0.848
*RAF1*	−0.011	0.384	−0.027	0.978
*KRAS*	0.198	0.305	0.649	0.519
*PIK3R1*	−0.009	0.492	−0.019	0.985
*PIK3C2G*	0.085	0.153	0.554	0.582
*ATM*	0.037	0.202	0.181	0.857
* **MAP2K4** *	0.730	0.339	2.153	**0.035**
*PIK3C2B*	−0.217	0.308	−0.704	0.484
*TGFBR2*	−0.360	0.342	−1.055	0.295
*SMAD2*	−0.768	0.656	−1.169	0.246
*CARD11*	0.009	0.179	0.052	0.959
*MAP3K1*	0.156	0.270	0.577	0.566
*SMAD4*	−0.523	0.429	−1.219	0.227
*MAP3K13*	−0.044	0.247	−0.177	0.860
*GNAS*	0.111	0.193	0.576	0.566
*JAK2*	0.236	0.158	1.498	0.139
*PTPN11*	−0.450	0.593	−0.759	0.450
*PPP2R2A*	−0.013	0.718	−0.017	0.986
*CASP8*	0.203	0.148	1.371	0.175
*BCL2L1*	0.522	0.532	0.982	0.330

SE, standard error. The significant genes were abstracted to generate a GMS (gene mutation score) = coefficient × gene mutation status (0: wild, 1: mut).

## Data Availability

The data that support the findings of this study are available upon reasonable request (e.g., for research purposes) from the authors.

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
