# Peer review of "The Prognosis Performance of a Neutrophil- and Lymphocyte-Associated Gene Mutation Score in a Head and Neck Cancer Cohort"

_biomedicines, 2023, doi:10.3390/biomedicines11123113_

Round 1

Reviewer 1 Report

Comments and Suggestions for Authors

In this study, Moi and colleagues proved the usefulness of the combination between neutrophil-to-lymphocyte ratio and gene mutation score to predict poor prognosis in head and neck squamous cell carcinomas. More than a hundred patients were rectrospectively examined and a rigorous statistical analysis was performed.

The paper is clear, written in good English and supported with useful tables and figures.

As a minor issue, I suggest to add a graphical abstract to attract the readers and explain the main findings of the work.

Reviewer 2 Report

Comments and Suggestions for Authors

In the article entitled “The prognosis performance of neutrophil and lymphocyte associated gene mutation score in head and neck cancer cohort”, the authors aimed to decipher the predictive value of the neutrophil-to-lymphocyte ratio (NLR) in patients with locally advanced head and neck squamous cell carcinomas (HNSCC) treated with chemoradiotherapy (CRT) and to explore the associations between NLR and the cancer genomic landscape. They collected clinical and next-generation sequencing (NGS) data with targeted sequencing information from 107 patients with HNSCC who underwent CRT and verified how patients with a high NLR belong to poor response group.

They also examined the genetic alterations linked to NLR and identified three candidate genes (MAP2K2, MAP2K4, and ABL1) that showed a significant correlation and were used to create a gene mutation score (GMS). They concluded that the high NLR-GMS group had significantly shorter relapse-free survival compared to the intermediate or low NLR-GMS group.

Data reported are potentially interesting and could be taken in consideration to support NLR as a promising clinical biomarker, and a prognostic marker in many solid tumors, including HNSCC.

However, the paper seems to be too preliminary and can’t be considered for publicaton in its present form and needs to be revised. I reported here my comments:

The authors should add an ABBREVIATIONS Section to the manuscript in order to clarify the meaning of such acronyms reported within the text.

In the RESULTS Section, paragraph named “3.1. Baseline characteristics of patients”, the authors reported that “All patients were diagnosed with HNSCC and underwent CRT”. What were the characteristics of CRT to which the patients underwent?” Did the patients underwent to the same treatments all? Have the authors analyzed untreated control samples? It would be interesting to compare data between CRT treated and normal samples in order to verify the “physiological status”, with respect to data CRT-treatment related.

In TABLE2, the authors reported “Pathways associated with lymphocytes and neutrophils derived using mutated gene in 199 HNSC cohort”. To what the authors refers when they reported the “Involved genes” column? Are there genes whose expression differs in CRT-treated with respect to untreated samples? Please clarify this data.

In the DISCUSSION Section, the authors reported that “To further evaluate the genetic alterations of the immune response associated with the NLR, we analyzed the NGS data of HNSCC patients and matched them with the KEGG Database”. The authors should add supplementary tables by reporting raw data from NGS analysis also adding the main pathways that resulted to be regulated in HNSCC.

It would be also interesting to validate the role of MAP2K2, MAP2K4, and ABL1 in CRT-mediated response of HNSCC patients. The authors should set some in vitro analysis by using related cell culture and functional assays.

I think that in this paper, the authors reported some potentially interesting data about the predictive value of the neutrophil-to-lymphocyte ratio (NLR) in patients with locally advanced head and neck squamous cell carcinomas (HNSCC) treated with chemoradiotherapy (CRT), suggesting it as a potential biomarker. The data are potentially interesting but too preliminary and need to be improved in order to be taken in consideration for publication.

The authors reported a possible correlation between HNSCC progression, CRT-mediated respone and MAP2K2, MAP2K4, and ABL1 expression, but it should be only a consequence (and not the cause) and cannot be evidenced in ALL samples. Futhermore, the analysis have to be done in normal control samples and normal cell lines to have a background data with respect to tumoral samples.

The paper also needs to be greatly improved in its content in order to meet the journal aims and be considered for publication.

Comments on the Quality of English Language

No comments

Author Response

In the article entitled “The prognosis performance of neutrophil and lymphocyte associated gene mutation score in head and neck cancer cohort”, the authors aimed to decipher the predictive value of the neutrophil-to-lymphocyte ratio (NLR) in patients with locally advanced head and neck squamous cell carcinomas (HNSCC) treated with chemoradiotherapy (CRT) and to explore the associations between NLR and the cancer genomic landscape. They collected clinical and next-generation sequencing (NGS) data with targeted sequencing information from 107 patients with HNSCC who underwent CRT and verified how patients with a high NLR belong to poor response group.

They also examined the genetic alterations linked to NLR and identified three candidate genes (MAP2K2, MAP2K4, and ABL1) that showed a significant correlation and were used to create a gene mutation score (GMS). They concluded that the high NLR-GMS group had significantly shorter relapse-free survival compared to the intermediate or low NLR-GMS group.

Data reported are potentially interesting and could be taken in consideration to support NLR as a promising clinical biomarker, and a prognostic marker in many solid tumors, including HNSCC.

However, the paper seems to be too preliminary and can’t be considered for publication in its present form and needs to be revised. I reported here my comments:

The authors should add an ABBREVIATIONS Section to the manuscript in order to clarify the meaning of such acronyms reported within the text.

  Thank you for your comment. We will add abbreviations below every tables in this article and try to avoid unnecessary acronyms.

In the RESULTS Section, paragraph named “3.1. Baseline characteristics of patients”, the authors reported that “All patients were diagnosed with HNSCC and underwent CRT”. What were the characteristics of CRT to which the patients underwent?” Did the patients underwent to the same treatments all? Have the authors analyzed untreated control samples? It would be interesting to compare data between CRT treated and normal samples in order to verify the “physiological status”, with respect to data CRT-treatment related.

  Thank you for your comment. All HNSCC patients in our study received CRT: 75 patients underwent CRT as postoperative adjuvant therapy and 32 patients received CRT as definitive treatment. The concurrent chemo-radiotherapy treatment included a total radiotherapy dose of 60–70 Gy and cisplatin-based chemotherapy. In the aspect of CRT, all patient received the same treatment and that’s the reason why we chose those patients for further genetic analysis in order to minimum the bias.

As for collecting patients without CRT as control samples, we are sorry that we didn’t analyze those patients without CRT. However, due to the importance role of CRT in HNSCC treatment, only few HNSCC patients could omit CRT during cancer treatment. However, it is still a very good suggestion that we may further investigate.   

In TABLE2, the authors reported “Pathways associated with lymphocytes and neutrophils derived using mutated gene in 199 HNSC cohort”. To what the authors refers when they reported the “Involved genes” column? Are there genes whose expression differs in CRT-treated with respect to untreated samples? Please clarify this data.

  Thank you for your comment. Table 2 reported the pathways associated with lymphocytes and neutrophils derived using mutated gene detected in 107 HNSC cohort. The “involved genes” column indicates the mutated gene found in our study cohort which are simultaneous involved in the correspond pathways. The “genes in pathways” column reported the overall genes number involved in the correspond pathways, and the “pathway percentage” was computed by dividing “involved genes” column and “genes in pathways” column, denoting the percentage of mutated genes found in study cohort involving in correspondence pathway.

In the DISCUSSION Section, the authors reported that “To further evaluate the genetic alterations of the immune response associated with the NLR, we analyzed the NGS data of HNSCC patients and matched them with the KEGG Database”. The authors should add supplementary tables by reporting raw data from NGS analysis also adding the main pathways that resulted to be regulated in HNSCC.

   Thank you for your comment. We have reported the NGS data for somatic mutation profiles of study cohort as Supplementary Table S1.

It would be also interesting to validate the role of MAP2K2, MAP2K4, and ABL1 in CRT-mediated response of HNSCC patients. The authors should set some in vitro analysis by using related cell culture and functional assays.

    Thank you for your comment. In fact, out next step is to perform in vitro experiments to confirm and validate the role of these three genes, MAP2K2, MAP2K4, and ABL1. However, we just started these experiments so preliminary data is not available at this point.

I think that in this paper, the authors reported some potentially interesting data about the predictive value of the neutrophil-to-lymphocyte ratio (NLR) in patients with locally advanced head and neck squamous cell carcinomas (HNSCC) treated with chemoradiotherapy (CRT), suggesting it as a potential biomarker. The data are potentially interesting but too preliminary and need to be improved in order to be taken in consideration for publication.

The authors reported a possible correlation between HNSCC progression, CRT-mediated response and MAP2K2, MAP2K4, and ABL1 expression, but it should be only a consequence (and not the cause) and cannot be evidenced in ALL samples. Furthermore, the analysis have to be done in normal control samples and normal cell lines to have a background data with respect to tumoral samples. The paper also needs to be greatly improved in its content in order to meet the journal aims and be considered for publication.

   Thank you for your thoughtful and constructive comments. The issue of NLR and the association with cancer prognosis have already been studied and documented for decades. However, no study had focused on the background genetic alternation behind NLR. Therefore, we came up with an idea. The first part of our plan was to collect valuable clinical and NGS data from patients and perform further analyses. We did find some target genes associated with NLR in patient with HNSCC underwent CRT. This finding is new and possess great potential. Therefore, we submit this article. We also start our second part of plan to confirm this finding by using in vitro experiment.

Reviewer 3 Report

Comments and Suggestions for Authors

In this investigation, Yeh et al. evaluated clinical and next-generation sequencing (NGS) data with targeted sequencing information from 107 patients with Head and neck squamous cell carcinomas (HNSCC) who underwent chemoradiotherapy (CRT). They described the difference in NLR between the good response group and the poor response groups as significant, with more patients having a high neutrophil-to-lymphocyte ratio (NLR) in the poor response group. Further, also examined the genetic alterations linked to NLR and found a total of 41 associated genes across eight common 36 pathways searched from the KEGG Database. Altogether, they showed high NLR-GMS (gene mutation score) group had significantly shorter relapse-free survival compared to other intermediate or lower scores. However, the authors need to address the following concerns.

1)    From Table 1, it was evident that only 7 females(6.5 %) among 107 patients were chosen in this study. Will these findings clearly hold the hypothesis proposed by the Authors? Including more females in cohorts gives more confidence levels.  

2)    In Table 1 “Cahracteristics” should be replaced with “Characteristics”

3)    In the selected cohort for baseline characteristics, what proportion of the patients exhibited inflammatory signs? Did the authors have considered any inflammatory signaling or pathway for this study?

4)    Represent all genes in italics throughout the manuscript.

5)     The p-values of AKT2 and PIK3CA in Table 4 are low. Why did the authors hadn’t-selected them as candidate genes along with MAP2K2, MAP2K4, and ABL1?

6)    Include information related to drugs that are used and currently available in clinics to target MAP2K2, MAP2K4, and ABL1 genes.

7)    The manuscript can be further revised for grammatical and typological errors.

Comments on the Quality of English Language

The manuscript can be further revised for grammatical and typological errors.

Author Response

Review 3

In this investigation, Yeh et al. evaluated clinical and next-generation sequencing (NGS) data with targeted sequencing information from 107 patients with Head and neck squamous cell carcinomas (HNSCC) who underwent chemoradiotherapy (CRT). They described the difference in NLR between the good response group and the poor response groups as significant, with more patients having a high neutrophil-to-lymphocyte ratio (NLR) in the poor response group. Further, also examined the genetic alterations linked to NLR and found a total of 41 associated genes across eight common 36 pathways searched from the KEGG Database. Altogether, they showed high NLR-GMS (gene mutation score) group had significantly shorter relapse-free survival compared to other intermediate or lower scores. However, the authors need to address the following concerns.

  • From Table 1, it was evident that only 7 females (6.5 %) among 107 patients were chosen in this study. Will these findings clearly hold the hypothesis proposed by the Authors? Including more females in cohorts gives more confidence levels.  

  Thank you for your comment. Our patients were all diagnosed with HNSCC. As we know, men are at twofold to fourfold higher risk than women for developing HNSCC (Nat Rev Dis Primers. 2020 Nov 26;6(1):92.). Males are much more susceptible to head and neck cancers than females regardless of whether they drink alcohol or smoke tobacco (Cancers (Basel). 2022 May 20;14(10):2521.). This gender difference is also obvious in Taiwan. Therefore, there were only 7 female patients in our cohort.

  • In Table 1 “Cahracteristics” should be replaced with “Characteristics”

  Thank you for your comment. We are sorry for this typo error and we replaced it with the right word.

  • In the selected cohort for baseline characteristics, what proportion of the patients exhibited inflammatory signs? Did the authors have considered any inflammatory signaling or pathway for this study?

  Thank you for your comment. Till now, several parameters have been reported to represent the degree of systemic inflammation, such as C-reactive protein, neutrophil-to-lymphocyte ratio (NLR), platelet-to-lymphocyte ratio, and lymphocyte-to-monocyte ratio. Due to NLR is the most promising clinical biomarker and has been shown to be prognostic in many solid tumors from previous studies, we chose NLR for further evaluation. Therefore, we only showed the data of absolute neutrophil count, absolute lymphocyte count and the NLR. Those data were retrieved from four weeks prior to commencing radiotherapy. If multiple values were available before treatment, those closest to the start date of radiotherapy were selected.

  • Represent all genes in italicsthroughout the manuscript.

  Thank you for your comment. We changed all genes in italics.

  • The p-values of AKT2 and PIK3CA in Table 4 are low. Why did the authors hadn’t select them as candidate genes along with MAP2K2, MAP2K4, and ABL1?

  Thank you for your comment. It’s true that the p-values of AKT2 and PIK3CA were also quite low (0.07 and 0.057), but they were still larger than 0.05. If the p-value is larger than 0.05, we cannot conclude that a significant difference exists. Therefore, we didn’t choose AKT2 and PIK3CA for further analysis.

  • Include information related to drugs that are used and currently available in clinics to target MAP2K2, MAP2K4, and ABL1 genes.

  Thank you for your comment. We had added the information related to drugs that target MAP2K2, MAP2K4, and ABL1 genes in the Discussion section.

  • The manuscript can be further revised for grammatical and typological errors.

Thank you for your comment. We will try our best to avoid grammatical or typological errors.

Round 2

Reviewer 2 Report

Comments and Suggestions for Authors

In the 2nd version of the article entitled “The prognosis performance of neutrophil and lymphocyte associated gene mutation score in head and neck cancer cohort”, the authors responded to my comments.

With respect to my comments, they almost completely responded to the arised questions.

Despite these considerations, I think that the manuscript needs to improve its in vitro experiments validating the role of MAP2K2, MAP2K4, and ABL1 in CRT-mediated response of HNSCC patients. Moreover, the authors should use related cell culture and functional assays to support their conclusions.

I think these data cannot be separated from those reported and presented in another paper since there could be not a real correlation and this paper could remain a speculative manuscript.

I confirm my first decision and ask for MAJOR REVISION in order to strongly support authors conclusions that I think are too preliminary to date.

Comments on the Quality of English Language

No comments

Round 3

Reviewer 2 Report

Comments and Suggestions for Authors

In this version of the article entitled “The prognosis performance of neutrophil and lymphocyte associated gene mutation score in head and neck cancer cohort”, the authors responded to my comments.

Regarding my considerations about the need to improve their in vitro experiments validating the role of MAP2K2, MAP2K4, and ABL1 in CRT-mediated response of HNSCC patients, the authors provided some interesting IHC data.

I think this data may represent the conclusion of this manuscript, but I suggest to continue these experiments in a future paper, in order to strongly support the role of MAP2K2, MAP2K4, and ABL1 in CRT-mediated response of HNSCC patients.

To conclude, I think the manuscript may be considered for publication in its present form.

Comments on the Quality of English Language

No comments

Author Response

Review 2 round 3

In this version of the article entitled “The prognosis performance of neutrophil and lymphocyte associated gene mutation score in head and neck cancer cohort”, the authors responded to my comments.

Regarding my considerations about the need to improve their in vitro experiments validating the role of MAP2K2, MAP2K4, and ABL1 in CRT-mediated response of HNSCC patients, the authors provided some interesting IHC data.

I think this data may represent the conclusion of this manuscript, but I suggest to continue these experiments in a future paper, in order to strongly support the role of MAP2K2, MAP2K4, and ABL1 in CRT-mediated response of HNSCC patients.

To conclude, I think the manuscript may be considered for publication in its present form.

Thank you for your comment. Your previous suggestions are really invaluable for us in enhancing the overall quality and impact of this article.